# Transcriptome Analysis Identifies Key Metabolic Changes in the Brain of *Takifugu rubripes* in Response to Chronic Hypoxia

**DOI:** 10.3390/genes13081347

**Published:** 2022-07-27

**Authors:** Fengqin Shang, Yun Lu, Yan Li, Bing Han, Renjie Wei, Shengmei Liu, Ying Liu, Yang Liu, Xiuli Wang

**Affiliations:** 1College of Fisheries and Life Science, Dalian Ocean University, Dalian 116023, China; fengqinshang96@gmail.com (F.S.); luyun0617@outlook.com (Y.L.); yanli10156@gmail.com (Y.L.); hanbing@dlou.edu.cn (B.H.); weirenjie0218@gmail.com (R.W.); liusm2200@sinovac.com (S.L.); 2College of Marine Technology and Environment, Dalian Ocean University, Dalian 116023, China; yingliu@dlou.edu.cn; 3Key Laboratory of Environment Controlled Aquaculture, Dalian Ocean University, Ministry of Education, Dalian 116023, China; 4Key Laboratory of Pufferfish Breeding and Culture in Liaoning Province, Dalian Ocean University, Dalian 116023, China

**Keywords:** chronic hypoxia, brain, transcriptome, *Takifugu rubripes*, gene expression, metabolic

## Abstract

The brain is considered to be an extremely sensitive tissue to hypoxia, and the brain of fish plays an important role in regulating growth and adapting to environmental changes. As an important aquatic organism in northern China, the economic yield of *Takifugu rubripes* is deeply influenced by the oxygen content of seawater. In this regard, we performed RNA-seq analysis of *T. rubripes* brains under hypoxia and normoxia to reveal the expression patterns of genes involved in the hypoxic response and their enrichment of metabolic pathways. Studies have shown that carbohydrate, lipid and amino acid metabolism are significant pathways for the enrichment of differentially expressed genes (DEGs) and that DEGs are significantly upregulated in those pathways. In addition, some biological processes such as the immune system and signal transduction, where enrichment is not significant but important, are also discussed. Interestingly, the DEGs associated with those pathways were significantly downregulated or inhibited. The present study reveals the mechanism of hypoxia tolerance in *T. rubripes* at the transcriptional level and provides a useful resource for studying the energy metabolism mechanism of hypoxia response in this species.

## 1. Introduction

Oxygen is an essential element for the survival of most organisms and a key substance involved in metabolism. However, changes in environmental, pathological and physiological conditions can lead to reduced oxygen availability [1]. Dissolved oxygen (DO) levels vary more dramatically in the aquatic environment than in the terrestrial environment [2]. Therefore, fish are better adapted to changes in DO levels than mammals and birds [3]. Generally, the DO content in the water body needs to be maintained above 4 mg/L for fish to grow normally. Once the DO concentration is equal to or below 2.3 mg/L, most fish start to show signs of breathing difficulties, such as floating head, which will lead to a decrease in growth rate and fertility, and when it is below 1 mg/L, most fish will float severely and die of suffocation [4]. In addition, The DO content in water bodies also affects the life processes of aquatic animals such as their metabolism and reproduction. The molecular mechanism of tolerance to hypoxia in aquatic animals is a basic scientific problem that has received wide attention. When fish are at low DO, their own biochemical and physiological status will also change, and these changes are reflected in growth, development, reproduction, behavior, metabolism, and antioxidant activities [5]. It has been reported that the main response of fish facing hypoxia is to maintain energy levels by improving the use of energy or reducing the consumption of energy [6]. For example, *Amphilophus trimaculatus* can cope with hypoxic conditions by reducing feeding to reduce metabolic energy consumption [7]. It was found in the brain and heart of *Danio rerio* and in the brain and liver of *Pelteobagrus vachelli* to partially compensate for cellular energy requirements by upregulating the expression of glycolytic enzymes such as phosphoglycerate kinase (PGK), phosphoglycerate mutase (PGAM) and hexokinase (HK) [8,9]. In addition, it has been shown that fish cells can also sense and transduce hypoxic stress signals and respond to hypoxic stress by modifying biochemical responses through gene expression regulation [10]. Complex biochemical reactions include decreased blood glucose concentration, inhibition of metabolic reactions such as aerobic oxidation of glucose, lactic acid accumulation, erythrocyte aggregation, increased affinity of hemoglobin for O_2_, and increased anaerobic respiration [11,12]. For example, under hypoxic conditions, the degradation of HIF-1α is terminated and accumulated, followed by the formation of heterodimers with HIF-1β in the nucleus, inducing its downstream hypoxic response genes, which in turn triggers a series of molecular response strategies, including oxygen sensor mobilization, oxygen transport, vascular and erythropoietin production [13].

Studies have shown that approximately 20% of the oxygen ingested by the body is used by the brain, and that this oxygen is primarily used to produce the ATP needed to maintain electrical signaling at synapses and action potentials [14]. In addition, brain tissue is extremely sensitive to hypoxia, and a few minutes of hypoxia can cause severe neurological damage to the brain and even neuronal death. Studies have shown that hypoxia causes severe acidosis in brain cells, resulting in impaired oxidative metabolism and inadequate energy supply, as well as reduced autoregulation of the cerebrovascular system, and narrowing or even occlusion of the lumen, resulting in irreversible damage to the brain parenchyma [15,16]. Cerebral hypoxia tends to cause increased vascular permeability in brain tissue and the accumulation of certain metabolites to form cerebral edema, while increased intracranial pressure and further reduction in cerebral blood flow can cause severe metabolic disorders in brain cells, which can eventually even lead to brain atrophy [15]. The vertebrate brain is one of the most metabolically active of all organs and is very sensitive to perturbations of energy metabolism [17]. Two unique conditions of the central nervous system (CNS) that distinguish the brain from other tissues are the inability of individual nerve cells to function autonomously and the blood–brain barrier (BBB) that selectively limits the rate of transfer of soluble substances between the blood and the brain. However, the biochemical pathways of energy metabolism in the brain are in most respects the same as in other tissues [17]. Nevertheless, not all animals suffer equally severe brain damage during hypoxia. Studies have shown that animals living in hypoxic environments for long periods of time have evolved unique and effective adaptation strategies to ensure their normal life under hypoxic conditions [18,19,20]. Mammals that specialize in diving rely on large amounts of glycogen and endogenous stores of O_2_ (e.g., bound to hemoglobin in the blood or to myoglobin in skeletal muscle) to support their own oxidative metabolic processes in the brain during diving [21,22].

*T. rubripes* belongs to the order Tetraodontiformers, suborder Tetraodontoidei, family Tetraodontidae, and genus *Takifugu*, which is mainly distributed in coastal China, Korean Peninsula and Japan, and is popular for its delicious meat and high nutritional value. Brenner et al. first proposed *T. rubripes* as a model organism for vertebrate genetic studies [23], and the first sequencing of the *T. rubripes* genome, which was only 400 Mb in size, was completed in 2002. With the significant expansion of *T. rubripes* aquaculture in recent years, the problem of hypoxia due to high density breeding, sudden power outages or other accidents has also arisen. Our previous studies on acute hypoxic stress (6 h) in *T. rubripes* showed that the brain of *T. rubripes* avoids brain damage by regulating its circadian rhythms and neurodevelopment, as well as increasing blood flow [24]. Although studies on the adaptation of *T. rubripes* to hypoxic stress have been a hot topic in recent years, there is still a lack of research on the energy metabolism of *T. rubripes* due to hypoxic stress in the aquatic environment. Therefore, this thesis focuses on the adaptive mechanisms of brain tissue in the *T. rubripes* under chronic hypoxic stress from the perspective of nutrient metabolism and energy utilization. The study of the mechanism of hypoxic response in fish brain not only helps us to understand the various pathways involved in regulating brain metabolism under hypoxic stress, but also provides new insights into the adaptive molecular mechanisms that arise when the brain responds to hypoxic stress.

## 2. Materials and Methods

### 2.1. Hypoxia Treatment and Fish Sampling

Animal experiments were approved by the Animal Care and Use committee at Dalian Ocean University. *T. rubripes* (body length: 21.7 ± 1.1 cm; body weight: 302 ± 52.5 g; a random mix of females and males) were obtained from Dalian Fugu Aquatic Co., in Dalian, Liaoning, China. The fish were domesticated in 300 L tanks with a flow-through seawater supply (DO was 7.5 ± 0.5 mg/L, 17 ± 0.5 °C) for one month before the start of the experiment. In the formal experiment, the DO in the anoxic experimental tank was reduced to 2.5 ± 0.5 mg/L by pumping nitrogen into the water through a nitrogen gas cylinder, and then 30 individual *T. rubripes* were randomly selected and placed in the normoxic experimental tank (DO was 7.5 ± 0.5 mg/L, Nor_BR) and anoxic experimental tank (DO was 2.5 ± 0.5 mg/L, Hyp_BR), respectively (15 fish for each tank). The experiment lasted for 10 days. In the experiments, the oxygen concentration was measured using an oxygen monitor (DO-Y100, Dissolved Oxygen Sensor, Wuhan, China). At the end of the experiment, twelve fish were randomly selected from the control and experimental groups, respectively. During sampling, the experimental fish were anesthetized using tricaine methanesulfonate (MS-222, Sigma, Redmond, WA, USA), and then their brain was dissected, immediately transferred into RNase-free tubes, quickly frozen in liquid nitrogen, and stored at −80 °C until further analysis.

### 2.2. Library Preparation and Illumina Sequencing

Total RNA was isolated from each sample using TRIzol reagent (Invitrogen, California, CA, USA) according to the manufacturer’s protocol. We used a NanoPhotometer spectrophotometer (IMPLEN, California, CA, USA) to detect RNA purity and an Agilent 2100 Bioanalyzer (Agilent Technologies, Santa Clara, CA, USA) for the precise detection of RNA integrity. Six sequencing libraries were finally constructed from the brain samples of 24 fish (three control groups and three experimental groups, each group with four biological replicates), and double-end (PE) sequencing was performed using the INovaseq 6000 (Illumina, San Diego, CA, USA).

### 2.3. RNA-Seq Quality Filtering and Mapping

The raw data were firstly quality controlled using FastQC (www.bioinformatics.babraham.ac.uk/projects/fastqc/, accessed on 10 March 2022). Trimmomatic v0.38 [25] was used to remove reads containing adapter, ploy-N and low-quality reads to get clean data, while Q20, Q30 and GC content calculations were performed on the clean data. All downstream analyses were performed based on the clean data with high quality. We downloaded the reference genome (assembly fTakRub1.2, www.ncbi.nlm.nih.gov/genome/63, accessed on 15 February 2022) of *T. rubripes* from NCBI (The National Center for Biotechnology Information, Bethesda, MD, USA). Clean reads were mapped to the reference genome using Hisat2 v2.1.0 [26]. HTSeq v0.11.2 was used to calculate the number of reads mapped to each gene [27]. The R package DESeq2 v3.10 [28] was used to analyze differentially expressed genes (DEGs), and an adjusted *p* value < 0.05 was used as a criterion for screening DEGs. The R package clusterProfiler v3.18.1 [29] was used for GO and KEGG enrichment analysis. The software Cytoscape v3.7.1 was used to map the enriched gene and KEGG signaling pathway network interactions [30]. Principal component analysis (PCA) was performed using an in-house python script, which was published in our previous article [24].

### 2.4. Experimental Validation by qRT-PCR

Quantitative real-time PCR (qRT-PCR) was performed with the extracted total RNA as described previously to confirm the accuracy of the transcriptome data. An amount of 1 µg of total RNA was reverse transcribed into cDNA using the HiScript^®^ II Q RT SuperMix for qPCR (Vazyme Biotech, Nanjing, China). Selected genes were amplified on the ABI StepOnePlus™ Real-Time PCR System (Life Technologies, Carlsbad, CA, USA) using the Green-2-Go 2 × qPCR-ROX Mastermix (Sangon Biotech, Shanghai, China). β-actin was selected as a reference to assess the expression levels of selected genes. Three biological replicates were performed for each treatment group, with three technical replicates for each biological replicate. Primers for each gene are listed in Appendix A. A melting curve analysis was performed, and the specificity of the amplification and fold-change value of selected genes was calculated using the 2^−ΔΔCt^ method [31].

## 3. Results

### 3.1. Illumina Sequencing and Reads Mapping

To identify the transcriptome response to chronic hypoxic stress in the brain of *T. rubripes*, six cDNA libraries (three experimental and three control groups, four samples per group) were generated for this experiment. A total of 445,613,754 raw reads were obtained using the Illumina NovaSeq 6000 sequencing platform. After discarding the low-quality raw reads, 433,452,502 clean reads were available. The scores of Q20 and Q30 levels exceeded 97 and 93%, respectively. The GC content in the sequence of each group was more than 49.64% and normally distributed (Appendix A). The clean reads of each group were sequenced and compared with the reference genome of *T. rubripes*. The alignment efficiency was more than 93.57% (Appendix A), and the analysis of mapping results by FastQC showed that the level of duplication in mapped reads was about 23.9–28.5% from all samples (Appendix A), indicating that the sequencing results were good and met the requirements of the subsequent analysis. All clean libraries’ sequencing data have been submitted to the NCBI Sequence Read Archive (SRA) database following accession number PRJNA645780.

### 3.2. Analysis of DEGs

The counted reads by HTSeq were loaded by DESeq2 for differential expression analysis (Appendix A). In total, 187 genes were identified as DEGs among the treatment groups using the criteria of adjusted *p* value < 0.05 (Appendix A). When compared with those of the control group, 131 genes were upregulated and 56 genes were downregulated under chronic hypoxic stress (Figure 1a). Interestingly, of those 187 DEGs, only 26 genes had a difference greater than two-fold (Figure 1b).

Notably, the expression of genes *aldoa*, *pfkfb4*, *ugt1a1*, *pnpla2*, *prxl2b*, *gcsh*, *aldh4a1*, and *aldh6a1*, which are related to carbohydrate, lipid, and amino acid metabolism, were upregulated (Figure 2a–h), whereas the expression of the genes *sema3e* and *plcb1*, which are related to the immune system and signal transduction, were downregulated (Figure 2i,j). However, *hif1* (and *hif1an*) did not show significant changes (Figure 2k,l).

To further explore the expression pattern of the DEGs, the raw output from HTSeq was normalized and subject to a variance-stabilizing transformation within DESeq2 (Appendix A). Clustering, heatmap and principal components analysis were performed following a variance-stabilizing transformation (Figure 3, Figure 4 and Figure 5). The images show that DEGs clustered the six groups of samples into two major branches and that genes in the same branch had similar expression patterns under different processing conditions. This indicates that chronic hypoxic stress has an effect on the brain of *T. rubripes* and that the biological replication between the same treatment groups was good.

### 3.3. Gene Ontology Analysis

To carry out an in-depth analysis of DEGs, we performed the GO enrichment analysis of the 187 DEGs (Figure 6). The GO analysis identified 3154 GO terms significantly enriched with the DEGs: 2449 biological process terms, 408 molecular function terms and 297 cellular component terms (Appendix A). For the biological processes enriched with the most DEGs (Figure 6a), cellular detoxification and organic substance metabolic processes were found to be the most dominant processes, including “detoxification”, “cellular oxidant detoxification”, “cellular detoxification”, “glucose catabolic process”, “glycolytic process through fructose-6-phosphate”, and “response to lipopolysaccharide”, indicating the importance of those biological processes in hypoxia adaptation. Furthermore, for cellular components (Figure 6b), we found that DEGs were predominantly enriched in “secretory granule lumen”, “cytoplasmic vesicle lumen”, “collagen-containing extracellular matrix”, “ficolin-1-rich granule”, and “ficolin-1-rich granule lumen” terms, suggesting that immune-related genes such as Ficolin-1 may be responsive to chronic hypoxic stress. For molecular function (Figure 6c), we found that most of the DEGs were enriched in terms related to the regulation of enzyme activity, indicating that hypoxic stress affects the level of protease activity in the organism. Interestingly, we also found that the significantly enriched GO terms could be well-clustered into eight categories (Figure 7), with the majority being metabolism-related categories, including “α-amino carboxylic organic molecule”, “glucose glycolytic hexose biosynthetic”, “glutamate glutamine family palmitoylation”, and “fatty by biotic differentiation”.

### 3.4. KEGG Analysis

Pathway analysis of DEGs identified between the control group and hypoxia-treated group based on KEGG database is shown in Appendix A. As shown in Figure 8, the metabolism-related pathways were significantly enriched and accounted for the most, including “Alanine, aspartate and glutamate metabolism”, “Carbon metabolism”, “Biosynthesis of amino acids”, and “Glycerolipid metabolism”. Interestingly, through the network diagram we found that the pathways and genes related to metabolism are well clustered (Figure 9). Among them, the genes *plcb1*, *plpp2*, *aldoa* and *aldh6a1* have a higher degree and connect the most network pathways, suggesting that these genes may play an important function under hypoxic stress. This was also confirmed in previous studies, where Cheng et al. showed that *plcb1* is an important hub gene and node in the hypoxic preconditioning signaling network [32]. Kawai et al. demonstrated that *aldoa* is a hypoxia-induced prognostic factor that is closely associated with the malignancy of colorectal cancer [33].

### 3.5. Validation of RNA-Seq Results by qRT-PCR

To validate our Illumina sequencing results, four genes were randomly chosen to validate the RNA-seq results (Figure 10). The qRT-PCR results generally agreed with the RNA-seq high-throughput sequencing data, indicating similar expression patterns of up- and downregulated genes in the RNA-seq and qRT-PCR tests.

## 4. Discussion

Hypoxia leads to changes in the physiological functions of cells and tissues, including energy metabolism, apoptosis, cellular defense, blood vessel formation and transport, cell proliferation, as well as other physiological and biochemical processes. It even triggers a series of diseases and injuries [34]. Changing metabolic pathways and metabolic patterns is one of the main strategies for long-term adaptation of fish to low oxygen environments. Our RNA-seq results provide more depth and coverage of the gene expression of *T. rubripes* at different DO concentrations (7.5 ± 0.5 and 2.5 ± 0.5 mg/L) for 10 days. The current study represents our first attempt to study the molecular response of *T. rubripes* to chronic hypoxia. We found that genes related to nutrient metabolism and energy utilization were highly significantly enriched and mostly upregulated due to chronic hypoxia, while genes related to immunity and signal transduction were not significantly enriched.

### 4.1. Effect of Hypoxia on Carbohydrate Metabolism in T. rubripes

Aldolase A (ALDOA) is a catalase that is closely related to glycolysis [35]. It has been shown that ALDOA can catalyze the glucose-driven glycolytic pathway of fructose-1,6-diphosphate to produce dihydroxyacetone phosphate (DHAP) and glyceraldehyde-3-phosphate (GAP) [36]. Interestingly, it was shown that *aldoa* is regulated by hypoxia-inducible factor-1alpha (HIF-1α) [37], a key regulator that controls hypoxia-induced transcription and metabolism. Hypoxia was shown to induce *aldoa* transcriptional upregulation and enhance glycolysis [38,39], which remains consistent with our results. Furthermore, in this study, we found that *aldoa* was upregulated not only in glucose-driven glycolysis, but also in fructose-driven glycolysis and the pentose phosphate pathway (PPP). Fructose-driven glycolytic metabolism can cross over the major rate-limiting step of phosphofructokinase (PFK) in glucose-driven glycolysis. Thus, fructose enters glycolysis faster than glucose and can rapidly supply energy to the body. Studies have shown that under hypoxic conditions, *Heterocephalus glaber* will survive for up to 18 min using fructose as an energy source to sustain the body’s vital activities. Park et al. also identified genes related to fructose-driven glycolysis that are highly expressed in skeletal muscle, heart, liver and brain [40]. Fructose-driven glycolysis could avoid the normal lethal effects of the body under hypoxia, and this mechanism could medically reduce hypoxic injury in human diseases. Hypoxia is a common factor that induces gene expression, especially in genes involved in maintaining cellular energy during glycolysis [41]. *Pfkfb4* is induced by hypoxia and is required for the hypoxia-induced glycolytic response [42]. It has been shown that one of the most important functions of *pfkfb4* in tumor cells is to regulate the metabolic flux of glycolysis and PPP through the allosteric regulation of glycolysis. In addition, *pfkfb4* plays a key role in managing reactive oxygen species (ROS) accumulation by transferring glucose metabolic intermediates to the PPP in various cancer cells [43,44]. Strohecker et al. also found that *pfkfb4* acts as a novel regulator of autophagy by modulating ROS levels through regulating PPP flux and NADPH production, thereby regulating autophagy [45]. Our results showed that *pfkfb4* expression was upregulated under chronic hypoxic stress. We reasonably speculate that the upregulated expression of *pfkfb4* may maintain cellular redox homeostasis by balancing glycolytic activity and antioxidant products, ultimately reducing oxidative stress and death of brain cells, and minimizing the damage of chronic hypoxia to the brain. UGTs enzyme is a glucuronide-based catalase that binds glucuronide groups to the hydroxyl, carboxyl, sulfhydryl, and amino groups of the drug to form the polar product glucuronide and excrete it from the body. Studies have shown that *ugt* plays a central role in the cellular detoxification of several compounds, preventing the accumulation of potentially dangerous exogenous substances or metabolites and thus their subsequent bioactivation into more toxic active intermediates [46,47]. Hypoxia can promote the production of such reactive substances, leading to an increased susceptibility to cellular damage [48]. Magnanti et al. found that *ugt1a* expression was significantly reduced in *Rattus norvegicus* peritubular myocytes under hypoxic conditions (1% O_2_), and hypothesized that due to the reduction of *ugt1a*, the excretion of some harmful substances was slowed down, which could aggravate the pathology of the urinary system or gonads [49]. It has been shown that the downregulated expression of *ugt1a1* under hypoxic stress slows down the metabolism of carcinogenic benzopyrene in humans, which in turn affects body homeostasis [50]. Here, we showed that *ugt1a1* was upregulated with decreasing DO concentration. An upregulated expression of *ugt1a1* may avoid damage to the organism from chronic hypoxia by accelerating the excretion of harmful substances from the brain. Collectively, under chronic hypoxic stress, *T. rubripes* may achieve rapid energy supply to the organism through a gradual shift from glucose to fructose metabolism. In addition, the organism actively maintains the redox balance of cells by balancing the glycolytic activity and antioxidant products to reduce the oxidative stress and death of brain cells.

### 4.2. Effect of Hypoxia on Lipids Metabolism of T. rubripes

In addition to carbohydrates, lipids are also important energy-supplying substances for organisms. Hypoxic stress can also have an effect on lipid metabolism. It has been shown that hypoxia in adipocytes induces and promotes lipolysis, increases the release of fatty acids (FA), and inhibits the uptake and utilization of FA. FA released from adipose tissue are transferred to mitochondria in skeletal muscle, heart, and liver tissues for oxidative catabolism to produce ATP for energy supply [51]. Interestingly, our results show that genes related to lipid metabolism are significantly upregulated under chronic hypoxic stress. It has been shown that adipose triglyceride hydrolase (ATGL), encoded by the *pnpla2* gene, is an important component of the lipolysis process and is the rate-limiting enzyme that initiates triglyceride catabolism [52,53]. It is able to catalyze the hydrolysis of triglycerides to glycerol and FA and plays an important role in the regulation of energy metabolism in the body [54]. In addition, studies have shown that *pnpla2* plays an important role in lipid metabolism in both adipose and non-adipose tissues. We thus hypothesized that *pnpla2’s* upregulated expression under hypoxic stress promoted the release of FA, which was consistent with previous studies. In addition, it was shown that *prxl2b* is a member of the thioredoxin superfamily, which was earlier identified in the brains of *Muroidea* and *Sus* and is presumed to play an important role in the central nervous system [55]. In the present study we found that chronic hypoxia activated the arachidonic acid (ARA) metabolic pathway and that the gene *prxl2b* was involved in ARA metabolism and upregulated its expression under chronic hypoxia. In mammalian studies, ARA and its metabolites have been found to play an important role in influencing adipocyte development [56,57], however, the mechanisms of ARA on lipid metabolism in fish have been less reported. In a study on *Synechogobius hasta*, Luo et al. found that lipoproteinase (LPL) in the liver decreased with increasing ARA levels [58]. Martins et al. found in a study of *Sparus aurata* that ARA decreased gene expression of the key transcription factor *pparα* and hormone-sensitive lipase (HSL) in lipid catabolism metabolism [59]. Therefore, it is reasonable to hypothesize that the upregulated expression of *prxl2b* and activation of the ARA metabolic pathway during chronic hypoxia play an important role in triglycerides’ hydrolysis and fatty acid β-oxidation.

### 4.3. Effect of Hypoxia on Amino Acid Metabolism in T. rubripes

It has been reported that regardless of whether scleractinian fishes migrate, protein is preferentially or eventually converted to carbohydrate when energy supplies are low [60]. Mommsen et al. found that protein becomes the primary fuel at the end of hypoxia when all other substrates are depleted [61]. Studies on *Cyprinus carpio* showed a significant decrease in free amino acid levels after 3 h of exposure to hypoxic stress [62]. Similarly, genes related to amino acid metabolism were found to be significantly upregulated in our study under chronic hypoxic stress. Both *aldh4a1* and *aldh6a1* are members of the aldehyde dehydrogenase (ALDH) family, which can use NAD or NADP as coenzymes to oxidize aldehydes to the corresponding carboxylic acids. It has been reported that *aldh6a1* is involved in the metabolism of pyruvate and the catabolism of valine, leucine and isoleucine [63]. *Aldh6a1*-encoded methylmalonate–semialdehyde dehydrogenase (MMSDH) catalyzes the oxidative decarboxylation of malonic acid and methylmalonic acid methyl semialdehyde to form acetyl and propionyl-CoA, which play a role in the valine and pyrimidine catabolic pathways [64]. Studies have shown that *aldh4a1* is widely expressed in the brain and widely present in a variety of cells, and that the pyrroline-5-carboxylate dehydrogenase it encodes, which is associated with proline degradation, catalyzes the conversion of pyrroline-5-carboxylate to glutamate [65], and this resulting glutamate is converted to α-ketoglutarate, which is transported from the mitochondria via the malate/alpha-ketoglutarate carrier into the cytoplasm [66]. In addition, it was shown that the hydrogen carrier protein (H-protein), an important component of the glycine cleavage system (GCS), is encoded by the gene *gcsh*, which is a hydrogen carrier protein that transfers the aminomethyl fraction generated by glycine decarboxylation to T-protein for further degradation. Defects in the GCS, a major physiological pathway for glycine degradation, are responsible for nonketotic hyperglycinemia (NKH) [67]. Our study showed that the expression of *gcsh* was significantly upregulated under hypoxic stress, and the upregulated expression of *gcsh* promoted the degradation of glycine. In summary, under chronic hypoxic stress, some glycogenic amino acids (such as alanine, glycine and serine) in the brain of *T. rubripes* are catabolized for organismal energy supply. The combination of amino acid catabolism and gluconeogenic pathways in the brain may be an adaptive mechanism for maintaining blood glucose levels and providing continuous energy to the body under hypoxic stress, similar to what was found in *Gillichthys mirabilis* [68]. However, the specific role of certain amino acids requires further studies to understand the amino acid dynamics in aquatic animals under hypoxic stress.

### 4.4. Immunity, Stress Response and Signal Transduction

Semaphorins are a group of secreted or membrane-bound signaling proteins that act as chemokines in neuronal axon guidance and cell migration [69]. Semaphorin 3E (*sema3e*), a member of the Semaphorins, was initially found to be involved in neuronal guidance [70] and formation of vascular patterns [71]. It was shown that the interaction of the ligand *sema3e* with its high-affinity receptor Plexin D1 (Sema3E/Plexin D1) inhibits the proliferation of vascular endothelial cells and thus angiogenesis [72]. In addition, Sema3E-Plexin-D1 signaling was shown to be involved in axon growth and guidance [73]. Our results showed that *sema3e* expression was downregulated as oxygen concentrations decreased, and it is hypothesized that the brain of *T. rubripes* may minimize brain damage by actively repairing damaged nerves and promoting angiogenesis. This is consistent with our previous findings on the *T. rubripes* brain under acute hypoxic stress [24]. Phospholipase CB1 (*plcb1*) acts as an initial G protein-coupled receptor coupled to the PLCL3 heterodimer, which in turn catalyzes the conversion of phosphatidylinositol 4,5-bisphosphate (PIP2) to inositol 1,4,5-trisphosphate (IP3) with diacylglycerol (DAG) [74]. It was shown that *plcb1* is involved in the signaling pathway that regulates the inflammatory response, and silencing *plcb1* gene expression using siRNA technology leads to increased levels of downstream inflammatory factors such as IF-1, IF-6, and IF-18 expression [75], which further activate the specific immune response system and cause damage to vascular endothelial cells and other cells [76]. Furthermore, our study shows that *plcb1* can be involved in signal transduction in addition to the signaling pathway of inflammatory response. A previous study reported that *plcb1*, located in the nucleus, is involved in extracellular signaling to the nucleus and that the metabolism of phosphatidylinositol in the nucleus plays an important role in the processes of mRNA precursor shearing, cell growth, proliferation, cell cycle and differentiation [77]. Poli et al. reported that *plcb1* is associated with cell proliferation in the K562 cell line by downregulating protein kinase alpha, while upregulating cell cycle protein 3 allowed the S-phase of cells to grow and inhibited cell proliferation [78]. Thus, we speculate that the downregulated expression of *plcb1* may cause inflammatory responses in fish, and in addition, may reduce energy consumption under hypoxic stress by preventing basic physiological and biochemical processes such as cell proliferation. In conclusion, the brain of *T. rubripes* is tolerant to hypoxia, but hypoxia can also cause some degree of damage to it.

### 4.5. Comparative Analysis of the Brains of T. rubripes under Acute Hypoxic Stress

Combined with our previous work [24], we elucidated the differences in the transcriptional profiles of *T. rubripes* brains under acute hypoxia (6 h) and chronic hypoxia (10 days) environments. The responses of *T. rubripes* to acute hypoxia and chronic hypoxia differed in organic matter metabolism, signal transduction pathways, and immune system. Under acute hypoxia, the organism minimizes brain damage caused by hypoxia mainly by regulating circadian rhythms, neurodevelopment, and stress responses such as promoting angiogenesis and increasing blood flow, while the process of using organic matter metabolism for energy supply was not significantly enriched. Interestingly, under chronic hypoxic stress, we found that the main mode of adaptation to hypoxia in *T. rubripes* was through glycolysis, lipid catabolism and amino acid metabolism for energy supply. In contrast, circadian rhythms, neurodevelopment, and some other signal transduction pathways were not significantly enriched. In conclusion, we conclude that in order to better cope with chronic hypoxic stress compared to acute hypoxia tolerance, *T. rubripes* fish altered their patterns of nutrient and energy metabolism.

## 5. Conclusions

The present study was designed to determine the effect of chronic hypoxia on *T. rubripes*. Taken together, these experiments confirmed that under chronic hypoxic stress, *T. rubripes* actively regulates changes in metabolic pathways to provide energy by promoting glycolysis, fatty acid β-oxidation and amino acid metabolism. In addition, the brain of *T. rubripes* actively repairs damaged nerves and promotes angiogenesis to minimize brain damage. Nonetheless, this study also suggests that chronic hypoxia may trigger inflammatory responses in the organism.

In conclusion, our present study demonstrates the molecular mechanisms of adaptation in the brain of *T. rubripes* under chronic hypoxic stress, focusing on the elucidation of altered metabolic pathways and metabolic patterns as one of the main strategies for adaptation to chronic hypoxic stress in the brain of *T. rubripes*. The present study helps to further complement the molecular mechanisms in the brain of *T. rubripes* in response to hypoxic stress. Moreover, to the best of our knowledge, the studies on hypoxia tolerance in the brain tissue of *T. rubripes* have been limited to acute hypoxia so far. Therefore, this work is also our first attempt at transcriptome analysis of *T. rubripes* brain under chronic hypoxic stress.

## Figures and Tables

**Figure 1 genes-13-01347-f001:**
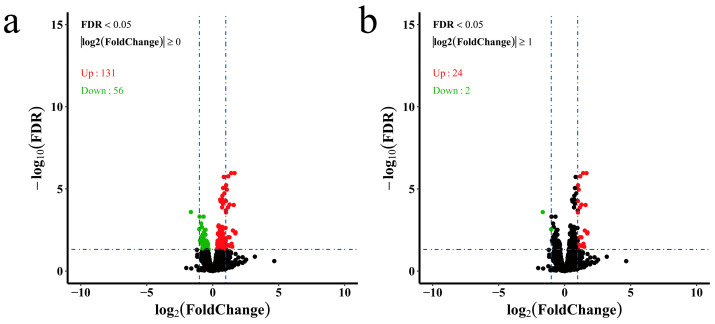
Volcano plot of differentially expressed genes. (**a**) DEGs with adjusted *p* value < 0.05 were chosen for volcano plot; (**b**) DEGs with adjusted *p* value < 0.05 and |log2FoldChange| ≥ 1 were chosen for volcano plot. The green dots represent downregulated genes, the red dots represent upregulated genes, and the black dots represent genes that were not significantly differentially expressed.

**Figure 2 genes-13-01347-f002:**
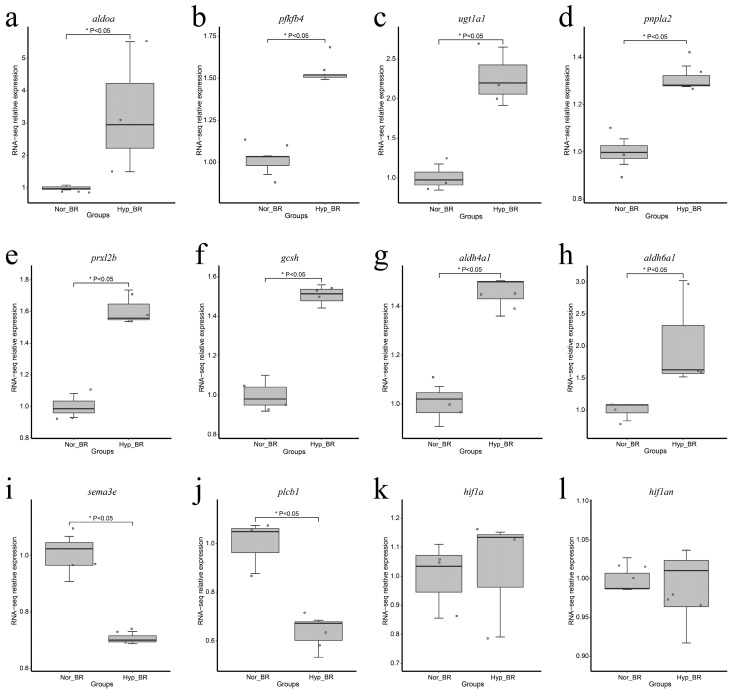
Boxplot shows gene expression at different dissolved oxygen concentrations. The solid line within boxplot represents median, while the boxplot bounds represent the 25 and 75% quantiles, and the dots represent the genes. * Represent P < 0.05 and P < 0.05 was considered statistically significant. The relative expression represents the fold change in expression of the target gene in the experimental group relative to the control group. (**a**) Boxplot for gene *aldoa*; (**b**) Boxplot for gene *pfkfb4*; (**c**) Boxplot for gene *ugt1a1*; (**d**) Boxplot for gene *pnpla2*; (**e**) Boxplot for gene *prxl2b*; (**f**) Boxplot for gene *gcsh*; (**g**) Boxplot for gene *aldh4a1*; (**h**) Boxplot for gene *aldh6a1*; (**i**) Boxplot for gene *sema3e*; (**j**) Boxplot for gene *plcb1*; (**k**) Boxplot for gene *hif1a*; (**l**) Boxplot for gene *hif1an*.

**Figure 3 genes-13-01347-f003:**
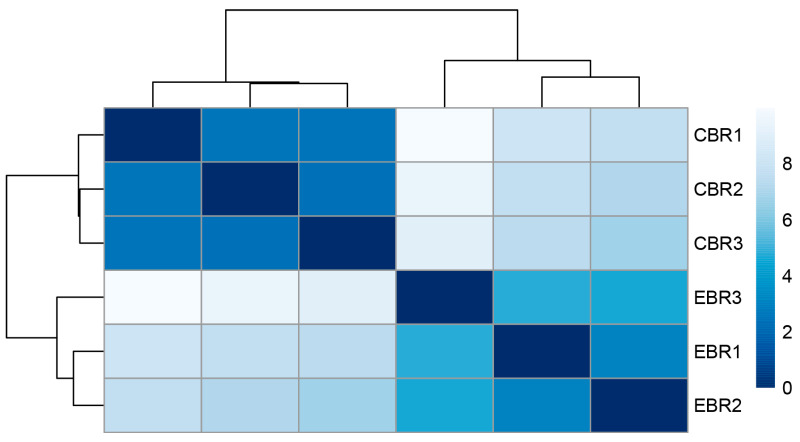
Heatmap of the clustering of samples from different treatment groups.

**Figure 4 genes-13-01347-f004:**
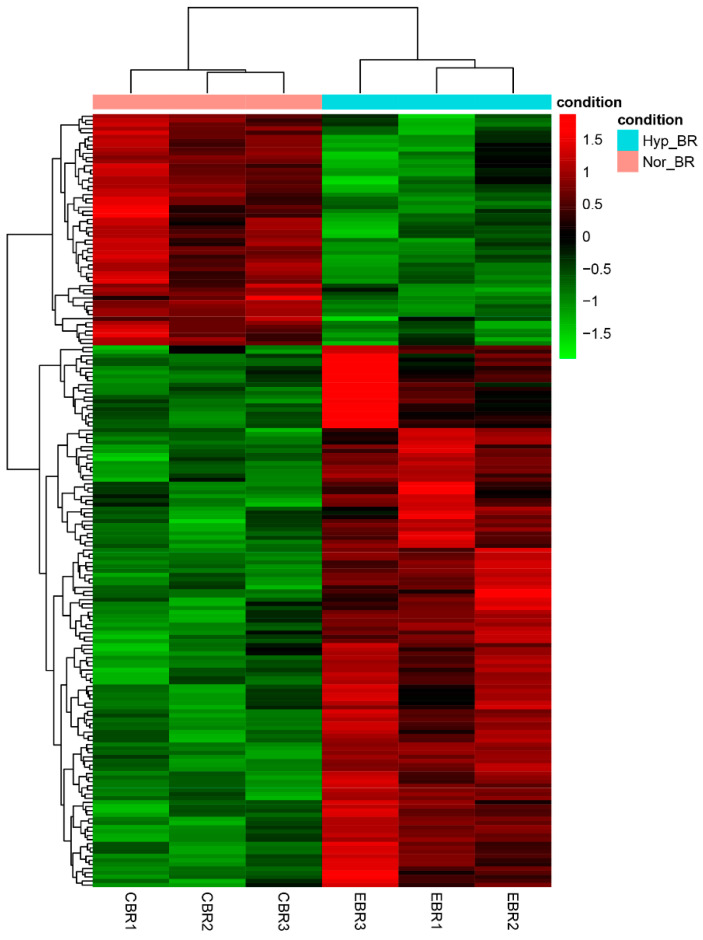
Heatmap of differentially expressed genes between the normoxic and hypoxic groups. Red represents upregulated genes and green indicates downregulated genes. Heatmap was drawn based on the normalized expression data (Appendix A).

**Figure 5 genes-13-01347-f005:**
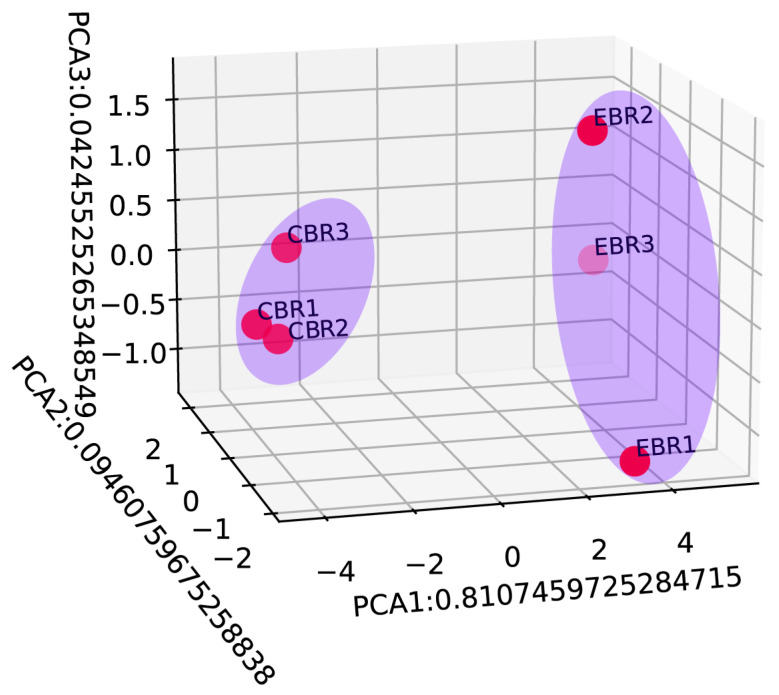
Principal component analysis (PCA) based on the normalized expression data of differentially expressed genes. All 6 samples were shaded by different light blue ellipses indicating the different treatment groups.

**Figure 6 genes-13-01347-f006:**
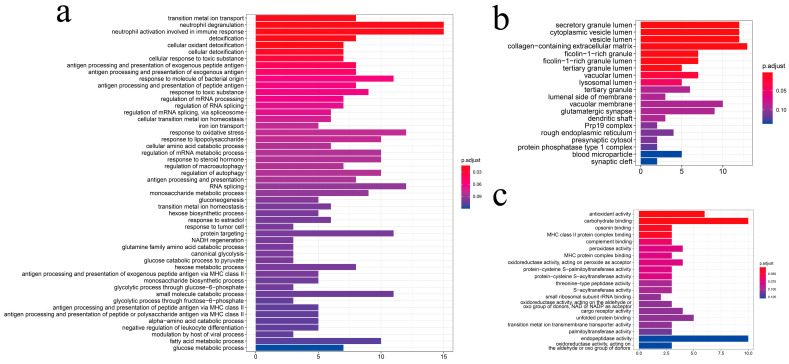
GO enrichment analysis of differentially expressed genes. (**a**) biological processes; (**b**) cellular components; (**c**) molecular functions.

**Figure 7 genes-13-01347-f007:**
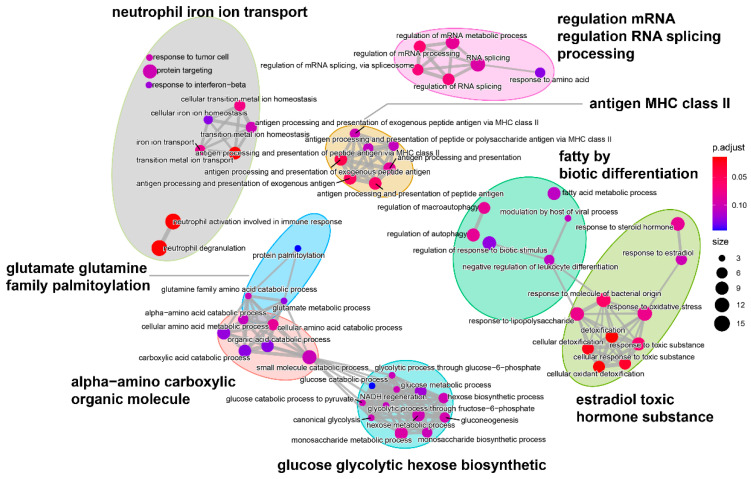
Functional grouping network diagram for GO enrichment analysis. The annotated GO terms were plotted in a network diagram, and the ellipses of different colors in the diagram represented clustered functional groups. Each point represented a GO term.

**Figure 8 genes-13-01347-f008:**
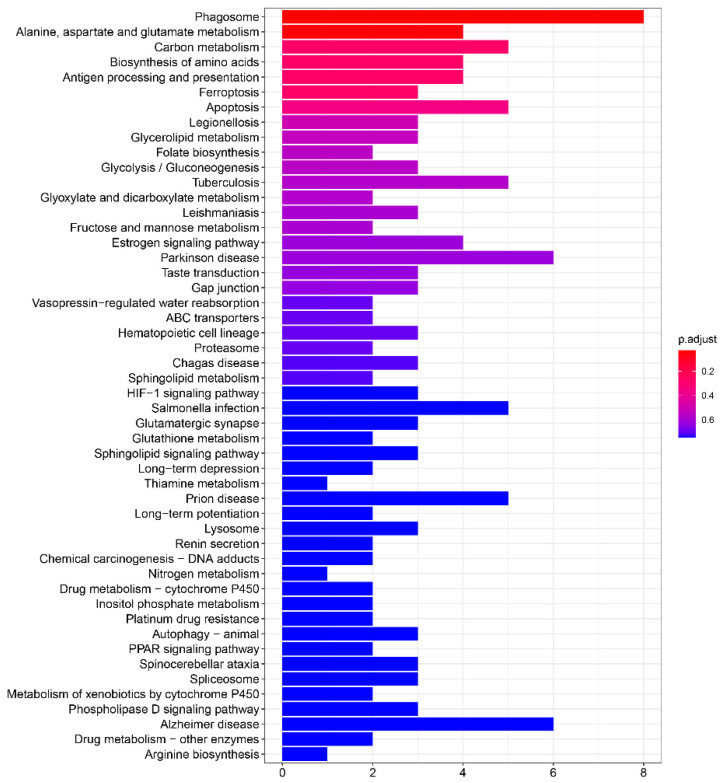
The top 50 enriched KEGG pathways.

**Figure 9 genes-13-01347-f009:**
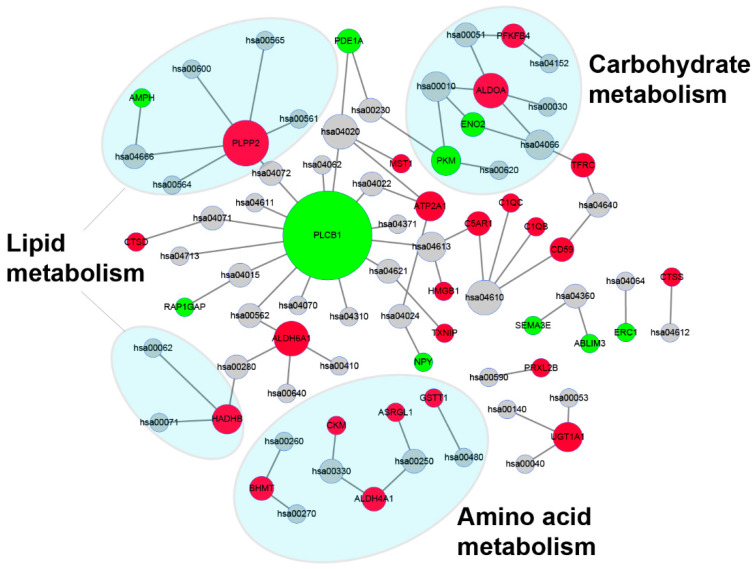
Network of enriched KEGG pathways and related genes. Red points represented the DEGs with an upregulated trend, green points are represented the DEGs with a downregulated trend, and gray points represent an enriched KEGG signaling pathway. Enriched metabolism-related pathways are shaded by a blue ellipse. Annotations of KEGG pathway are in Appendix A.

**Figure 10 genes-13-01347-f010:**
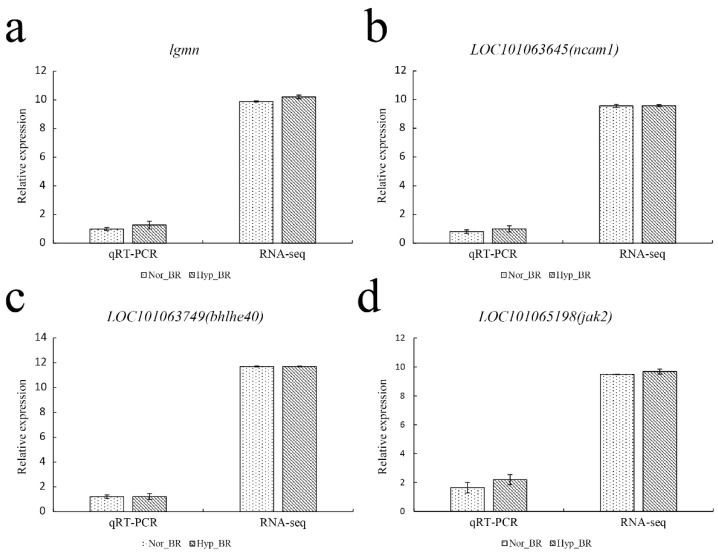
Quantitative real-time PCR verification. The gene expression levels were evaluated relative to the expression level of β-actin using the 2^−ΔΔCT^ method. The correlation between normalized counts from RNA-seq and the relative expression from qPCR was calculated.

## Data Availability

All sequencing data were submitted to the Sequence Read Archive (SRA) public database in NCBI (www.ncbi.nlm.nih.gov/, accessed on 13 December 2021), under the accession code PRJNA645780. All other data included in this study are available upon request by contact with the corresponding author (Yang Liu).

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
