# Peer review of "Transcriptome Analysis Identifies Key Metabolic Changes in the Brain of Takifugu rubripes in Response to Chronic Hypoxia"

_genes, 2022, doi:10.3390/genes13081347_

Round 1

Reviewer 1 Report

Summary of the research and overall impression

The authors present a study on RNAseq data of Takifugu rubripes housed under hypoxic and normoxic conditions. The underlying hypothesis is interesting and the subject of research generally interesting. However, the study shows drawbacks, as the validation of the RNAseq results has not been extensively undertaken.

Specific areas of improvement

-Major issues:

            In Figure 3, the gene expression changes between hypoxia and normoxia do not seem to show differentially regulated gene signatures but a very consistent picture between Normoxia and Hypoxia which does not correspond well with Figure 4, where there are wide changes between the PCA analysis of the different samples. This should be better reflected in Figure 3.

            Only randomly selected targets have been validated by qPCR expression analysis. While those seem to be very reliable, it would be desirable to validate the expression of the genes you are discussing in your Discussion

            No validation of protein levels has been undertaken. This is a major drawback, as protein turnover is a major characteristic feature of HIF stabilization that is not reflected by the RNA expression levels.

            The discussion may be overstating the findings, as there are a lot of speculative findings named. 

-Minor issues:

            The sex of the fishes chosen for the analysis should be noted, as hypoxia can differentially affect the different sexes.

Reviewer 2 Report

Interesting, some points need a revision:

- Lines 102-105: "The study of hypoxic response mechanism of fish will not.. hypoxia-tolerant fish species." But what is the aim of this paper?

- Lines 218-221: "For molecular function (Figure 5c), terms related to enzyme activity... to chronic hypoxic processes" It is not clear what authors want to say. Revise.

- Figure 3 is hard to read. Improve its figure legend.

- Lines 251-525: "It indicates that these genes may play an important function under hypoxic stress. " Then discuss more it.

- Lines 381-383: " In addition, studies on Acipenser schrenckii showed that... related to amino acid metabolism [62]. Is there any relationship with the Takifugu rubripes? Discuss

- Discussion section is very long and should be reduced.

- In the conclusion section, please add what this paper add new to the literature.

Author Response

Dear Assistant Editor and Reviewer,

Thank you for your letter and for the reviewers’ comments concerning our manuscript entitled “Transcriptome analysis identifies key metabolic changes in the brain of Takifugu rubripes in response to chronic hypoxia” (Manuscript ID: genes-1827155). Those comments are all valuable and very helpful for revising and improving our paper, as well as the important guiding significance to our research. We have studied comments carefully and have made the corrections which we hope meet with approval. The revised portion is marked in red in the paper. We hope that the revision is acceptable, and your favorable consideration of our manuscript is greatly appreciated.

The main corrections in the paper and the responses to the reviewer’s comments are as following:

Point 1: Lines 102-105: "The study of hypoxic response mechanism of fish will not.. hypoxia-tolerant fish species." But what is the aim of this paper?

Response 1: Thank you very much for your suggestion. We are sorry for our inappropriate expression, and we have changed the sentence " The study of hypoxic response mechanism of fish will not only help us to understand the evolution of hypoxic signaling pathways, but also has important implications for the selection and breeding of hypoxia-tolerant fish species." to " The study of the mechanism of hypoxic response in fish brain not only helps us to understand the various pathways involved in regulating brain metabolism under hypoxic stress, but also provides new insights into the adaptive molecular mechanisms that arise when the brain responds to hypoxic stress.".

Point 2: Lines 218-221: "For molecular function (Figure 5c), terms related to enzyme activity... to chronic hypoxic processes" It is not clear what authors want to say. Revise.

Response 2: Thank you very much for your suggestion. In the revised manuscript we have rewritten the sentences to “For molecular function (Figure 6c), we found that most of the DEGs were enriched to terms related to the regulation of enzyme activity, indicating that hypoxic stress affects the level of protease activity in the organism”.

Point 3: Figure 3 is hard to read. Improve its figure legend.

Response 3: Thank you very much for your suggestion. As suggested by the reviewers, we have changed the legend to “Heatmap of differentially expressed genes between the normoxic and hypoxic groups. Red represents upregulated genes and green indicates downregulated genes. Heatmap was drawn based on the normalized expression data (Supplementary Information Table S10)” in the revised manuscript.

Point 4: Lines 251-525: "It indicates that these genes may play an important function under hypoxic stress. " Then discuss more it.

Response 4: Thank you very much for your suggestion. Based on the reviewers' suggestions, we have added more discussion to the revised version of the manuscript. For example, by citing previous research results to better support our conclusions. The revised version is as follows: Among them, the genes plcb1, plpp2, aldoa and aldh6a1 have a higher degree and connect the most network pathways, suggesting that these genes may play an important function under hypoxic stress. This was also confirmed in previous studies, where cheng et al. showed that plcb1 is an important hub gene and node in the hypoxic preconditioning signaling network [32]. Kawai et al. demonstrated that aldoa is a hypoxia-induced prognostic factor that is closely associated with the malignancy of colorectal cancer [33]..

Point 5: Lines 381-383: " In addition, studies on Acipenser schrenckii showed that... related to amino acid metabolism [62]. Is there any relationship with the Takifugu rubripes? Discuss.

Response 5: Thank you very much for your suggestion. In this section we originally wanted to express that hypoxic stress significantly increased the expression of genes related to amino acid metabolism in Acipenser schrenckii, such as gene cyp1a. This is consistent with the significant upregulation of expression of genes related to amino acid metabolism in our current study in Takifugu rubripes under hypoxic stress. According to the reviewer's suggestion, we tried to further discuss the expression of gene cyp1a under different oxygen concentrations in this study in the revised manuscript, and we regret to find that gene cyp1a did not show significant differential expression in this study. Considering the rigor of the manuscript and the length of the discussion section, we have removed “In addition, studies on Acipenser schrenckii showed that hypoxic stress significantly increased the expression of cyp1a, a gene related to amino acid metabolism [62].” from the revised version of the manuscript.

Point 6: Discussion section is very long and should be reduced.

Response 6: Thank you very much for your suggestion. Based on your suggestions and those of other reviewers, we have removed inappropriate expressions and made changes in the Discussion section, and the length of the Discussion section has been shortened.

Point 7: In the conclusion section, please add what this paper add new to the literature.

Response 7: Thank you very much for your suggestions. As suggested by the reviewer, we have added new elements of this study to the literature in the revised manuscript. The additions are as follows: “In conclusion, our present study demonstrates the molecular mechanisms of adaptation in the brain of T. rubripes under chronic hypoxic stress, focusing on the elucidation of altered metabolic pathways and metabolic patterns as one of the main strategies for adaptation to chronic hypoxic stress in the brain of T. rubripes. The present study helps to further complement the molecular mechanisms in the brain of T. rubripes in response to hypoxic stress. Moreover, to the best of our knowledge, the studies on hypoxia tolerance in brain tissue of T. rubripes have been limited to acute hypoxia so far. Therefore, this work is also our first attempt at transcriptome analysis of T. rubripes brain under chronic hypoxic stress”. This has been marked in red in the revised manuscript.

We tried our best to improve the manuscript and made some changes in the manuscript. These changes will not influence the content and framework of the paper. Here, we do not list all the changes, but they are marked in red in the revised manuscript.

We appreciate for Editor/Reviewers’ warm work earnestly and hope that the correction will meet with approval.

Once again, thank you very much for your comments and suggestions.

Sincerely yours,

Ms. Fengqin Shang

College of Fisheries and Life Science,

Dalian Ocean University

52 Heishijiao Street, Dalian, 116023, Liaoning, China

Email address: fengqinshang96@gmail.com

Reviewer 3 Report

I checked your manuscript and described comments below.

I think this paper is a good analysis of the transcriptome response chronic hypoxia stress in the brain of T. rubripes. However supplementary files have not been uploaded.

It’s an important problem because I can’t see the data.

You should upload supplementary files.

Author Response

Response to Reviewer 3 Comments

Dear Assistant Editor and reviewer,

  Thank you for your letter and for the reviewers’ comments concerning our manuscript entitled “Transcriptome analysis identifies key metabolic changes in the brain of Takifugu rubripes in response to chronic hypoxia” (Manuscript ID: genes-1827155). We are very sorry that we mistakenly put the supplemental files in the manuscript.zip along with the manuscript when we uploaded the manuscript into the system. Now we tried to re-upload the supplemental file into the response to reviewers screen, but it failed. Please look in the manuscript.zip for the excel sheet with the file name: SupplementaryInformationS1-S10.xlsx. If you cannot find it, please do not hesitate to let us know and we will contact the editor to re-upload it. Again, we apologize for our oversight, and thank you very much for your kind comments on our manuscript.

  We appreciate for Editor/Reviewers’ warm work earnestly and hope that the correction will meet with approval.

Sincerely yours,

Ms. Fengqin Shang

College of Fisheries and Life Science,

Dalian Ocean University

52 Heishijiao Street, Dalian, 116023, Liaoning, China

Email address: fengqinshang96@gmail.com

Round 2

Reviewer 1 Report

The previous comments have not been sufficiently addressed. 

Reviewer 3 Report

I checked your manuscript and described comments below.

I have confirmed the contents of supplementary information S1-S10.

I think “Table S9” should also gene name. The gene symbol doesn't understand the contents well (i.e. ina, internexin neuronal intermediate filament protein alpha).

I don't think this paper has any major mistakes or grammatical problems about the modified part.
